# The Current State of Extracellular Matrix Therapy for Ischemic Heart Disease

**DOI:** 10.3390/medsci12010008

**Published:** 2024-01-29

**Authors:** Khaled Hamsho, Mark Broadwin, Christopher R. Stone, Frank W. Sellke, M. Ruhul Abid

**Affiliations:** 1Division of Cardiothoracic Surgery, Department of Surgery, Cardiovascular Research Center, Rhode Island Hospital, Alpert Medical School of Brown University, Providence, RI 02903, USA; khamsho@alfaisal.edu (K.H.); mbroadwin@lifespan.org (M.B.); christopher_stone@brown.edu (C.R.S.); fsellke@lifespan.org (F.W.S.); 2College of Medicine, Alfaisal University, Riyadh 11533, Saudi Arabia

**Keywords:** extracellular matrix, myocardial ischemia, ischemic heart disease, decellularized, tissue-derived ECM, ECM components, bioengineered ECM

## Abstract

The extracellular matrix (ECM) is a three-dimensional, acellular network of diverse structural and nonstructural proteins embedded within a gel-like ground substance composed of glycosaminoglycans and proteoglycans. The ECM serves numerous roles that vary according to the tissue in which it is situated. In the myocardium, the ECM acts as a collagen-based scaffold that mediates the transmission of contractile signals, provides means for paracrine signaling, and maintains nutritional and immunologic homeostasis. Given this spectrum, it is unsurprising that both the composition and role of the ECM has been found to be modulated in the context of cardiac pathology. Myocardial infarction (MI) provides a familiar example of this; the ECM changes in a way that is characteristic of the progressive phases of post-infarction healing. In recent years, this involvement in infarct pathophysiology has prompted a search for therapeutic targets: if ECM components facilitate healing, then their manipulation may accelerate recovery, or even reverse pre-existing damage. This possibility has been the subject of numerous efforts involving the integration of ECM-based therapies, either derived directly from biologic sources or bioengineered sources, into models of myocardial disease. In this paper, we provide a thorough review of the published literature on the use of the ECM as a novel therapy for ischemic heart disease, with a focus on biologically derived models, of both the whole ECM and the components thereof.

## 1. Extracellular Matrix (ECM)

The extracellular matrix (ECM) is a complex, three-dimensional network composed of structural and nonstructural proteins, glycosaminoglycans (GAGs), and proteoglycans (PGs), all of which communicate to maintain and support the structural and functional integrity of the tissue in which it is situated [1]. Accordingly, the ECM acts concomitantly as a structural scaffold for cellular organization, and a functional a conduit for intercellular communication, with components acting as signaling molecules and mediators of pathways that maintain tissue organization and participate in its remodeling [1,2]. While the constituents and roles of the ECM in all tissues exhibit many similarities, there are structural, architectural, biochemical, and functional details that are tissue-specific. For instance, the collagen-based ECM in skeletal muscle is arranged in a three-dimensional, tripartite hierarchical structure (the epimysium, perimysium, and the endomysium) that contributes to tension generation, which is an integral component for the optimal function of the muscle tissue [3]. In the liver, the ECM is notable for its capacity to foster hepatic regeneration and to structurally support the cellular, vascular, and ductal systems it embeds [4]. In all cases, the ECM is not an inert structure, but rather a dynamic network subject to contextual change. In the context of pathology, such changes are often driven by the immune system as part of the inflammatory and fibrotic responses to cellular insults [2,5].

## 2. Cardiac ECM in Health and in Ischemic Disease

In the heart, the myocardial ECM consists predominantly of a network of fibrillar collagens—85% type I and 11% type III—arranged in a hierarchal manner as the epimysium, the perimysium, and the endomysium in an analogous manner to that seen in skeletal muscle (Figure 1) [6]. The epimysium surrounds the entire myocardium and connects it to endothelial surfaces of the endocardium and epicardium; the perimysium arises as an extension of the epimysium and surrounds and interconnects individual muscle fibers; and the endomysium serves as the final extension of the network at the level of cardiomyocytes (CMs), supporting the microvasculature, nutrient exchange, and communication between the cells’ cytoskeletal network and the interstitium [7,8]. All of these ECM components are embedded within a gel-like substance consisting of different forms of GAGs and PGs [9], with other components including laminins, matricellular proteins, fibronectin, and elastin [7,10]. The production and remodeling of the cardiac ECM is mediated primarily by cardiac fibroblasts (CFs) [7,11]. Working together, these components function to transmit and distribute the contractile forces of the myocardium, exchange signals between cardiomyocytes, and provide a framework for multiple myocardial cellular populations [12].

In the post-myocardial infarction state, a variety of ECM-based myocardial changes characteristic of the triphasic cardiac response to injury occur: inflammation, proliferation, and maturation. First, the ECM is broken down by the ischemia-precipitated inflammatory activation of degradative enzymes such as the matrix metalloproteases (MMPs) [7]. This is followed by the generation of a provisional fibrin-based matrix derived from extravasated plasma proteins such as fibrinogen and fibronectin. While the exact role of the provisional matrix is not well understood, it is thought to provide structural support to proliferating cells and resident fibroblasts, and to foster the repair process through interactions with cell-surface integrins [7,13]. This has been borne out experimentally; the monoclonal antibody-mediated inhibition of dermal fibroblast integrins produces the inhibition of fibroblast attachment to fibrinogen and fibrin in the provisional matrix [14]. The transmigration of fibroblasts from a collagen-based matrix site to a fibrin-based provisional matrix site appears to be dependent on the presence of fibronectin. This was demonstrated in vitro by the decreased infiltration of fibroblasts into fibrin-based gel after the inhibition of the interaction between fibronectin and fibroblast integrin receptors [15]. Thereafter, in the proliferative phase, anti-inflammatory signals become predominant. The clearance of the plasma-derived matrix occurs and a second-order, cell-derived provisional matrix, primarily consisting of the fibronectin alternative splice variant extradomain-A (ED-A), hyaluronan, and versican, forms. This matrix is also enriched with matricellular proteins including periostin, osteopontin (OPN), thrombospondins, tenascin-C, SPARC, and CCN2, which contribute to cardiac repair response within the infarct and at its borders [7,13,16]. It is also understood to drive activation and differentiation of fibroblasts into myofibroblasts through an array of interactions with ED-A fibronectin, TGF-beta, matricellular proteins, and integrins. As the cells responsible for laying down the structural proteins of the ECM, myofibroblasts are believed to be the primary contributor to the repair process. Angiogenesis also occurs during this phase, driven by growth factors and repair signals such as vascular endothelial growth factor (VEGF) A, placental growth factor (PGF), fibroblast growth factor (FGF) 2, and Wnt family member 1 [17]. The final maturation phase is characterized by ECM cross-linking. Myofibroblasts appear to enter a quiescent state and eventually undergo apoptosis, and the formation of a mature scar that replaces the infarct is completed [7,13]. After large infarcts, chronic inflammatory signaling and interstitial fibrosis and accompanying structural changes in the myocardium result in increased wall stress that eventually contributes to heart failure [18,19,20].

Additionally, the cellular actions of immune cells including neutrophils, monocytes, dendritic cells, and lymphocytes are implicated in modulating cardiac fibrosis post-MI. Innate immune cells such as neutrophils and macrophages initially contribute to the remodeling process in the first weeks following MI [21]. Macrophages contribute to fibrosis by directly transforming into a fibroblast-like phenotype and by driving myofibroblast transdifferentiation by secreting TGF-beta [21,22,23]. Immune cells are also implicated in chronic remodeling and progression to heart failure. For instance, dendritic cells and monocytes contribute to non-resolving chronic inflammation following MI, which ultimately contributes to the progression of heart failure [24]. Furthermore, the activation of CD4+ and CD8+ T cells is necessary to drive the progression of ischemic heart failure, suggesting that the adaptive immune system further contributes to the chronic inflammation seen in cardiac remodeling [25]. Dysfunctional Tregs, exhibiting pro-inflammatory and antiangiogenic properties, also contribute to the pathological remodeling seen in chronic ischemic cardiac failure, evidenced by reduced fibrosis and enhanced neovascularization following the selective ablation of these cells in mice [26]. In fact, CD4+ cells demonstrate a biphasic response post-MI. A controlled response of CD4+ cells promote wound-healing and scar formation and then promptly recess after the cessation of the acute inflammatory response post-MI. However, in chronic ischemic heart failure, another surge of CD4+ cells target the heart and promote pathological remodeling [27]. The exact mechanism by which this secondary response occurs is not well-understood.

## 3. ECM in the Treatment of Ischemic Heart Disease

Given the changes that the myocardium manifests in response to ischemia, the ECM provides numerous therapeutic targets for the amelioration of chronic remodeling and progression to heart failure. Promising targets include coronary angiogenesis, stem cell recruitment and differentiation, ventricular remodeling, and inflammatory or fibrotic cell recruitment. Currently, the use of the ECM as a treatment for ischemic heart disease (IHD) falls under three broad categories: (1) individual components extracted from the ECM; (2) tissue-derived ECM; and (3) bioengineered ECM. In this section, we present different types of ECM-based therapeutic models that have been used in animal IHD models, with a focus on their benefits and potential therapeutic efficacy. A schematic summary of these different models using the ECM is shown in Figure 2. The application of ECM-based therapy and desired outcomes following treatment are shown in Figure 3.

## 4. Individual ECM Components

This category includes numerous macromolecular ECM constituents, including collagen, fibrin, hyaluronic acid, periostin, agrin, fibronectin, SLIT2, and nephronectin, as shown in Figure 1. Below, we discuss each major component of the ECM and underscore the principal features and effects of these as potential therapies for IHD. We also provide a helpful summary of the recently published literature that reported ECM use in animal IHD models with their major findings (Table 1).

### 4.1. Collagen

Collagens make up the largest component of the ECM in cardiac tissue. Given their structural ubiquity, treatments for IHD have primarily employed collagen as a foundational biomaterial for combination with other treatment modalities, rather than as a stand-alone treatment [40]. The generation of cardiac tissue grafts by seeding cells on collagen scaffolds [41] and the production of cardiac patches with collagen as a component of the scaffold material [42] are included among such combinatorial strategies. In addition to this, however, several studies have attempted to employ collagen as a standalone treatment [28,29,43,44,45].

Dai et al. conducted an experiment that used collagen to thicken the left ventricular wall of infarcted rat hearts in an effort to prevent adverse post-infarct structural remodeling [28]. Hemodynamics, wall thickness, infarct size, regional myocardial blood flow, and post-mortem LV volumes were among the parameters characterized, with improvement noted in ejection fraction (EF), stroke volume (SV), myocardial scar thickness, and paradoxical systolic bulging. Similarly, McLaughlin et al. explored the effects of injectable recombinant collagens I (rHCI) and III (rHCIII) in a mouse MI model [29]: mice that received administrations of rHCI exhibited enhanced EF, which increased from 34.6% post-MI to 44.2% 28 days post-treatment. Conversely, rHCIII-treated mice maintained their baseline EF, while saline injection control mice experienced EF deterioration at 28 days post-injection. Similarly, the rHCI and rHCIII groups demonstrated reduced end-systolic volume changes compared to control animals at 28 days, along with an improved SV and cardiac output. Additionally, scar size as a percentage of the left ventricle (LV) was smaller in the rHCI (31%) and rHCIII (37%) groups compared to the control group (52%), and rHCI produced increased myocardial wall thickness (1.3 mm) outside the infarct zone compared to the control value (0.96 mm). Structurally, increased capillary density at the infarct border zone, along with higher troponin I and connexin 43 expression, was observed in rHCI and rHCIII groups, indicating a pro-angiogenic effect of the injected collagens. Finally, the treatment appeared to modulate the inflammatory profile of infarcted mice, with more anti-inflammatory M2 macrophages observed in the scar and reduced recruitment of bone marrow-derived inflammatory cells and leukocytes post-MI with rHCI treatment. In vitro experiments corroborated these findings, demonstrating greater M2 macrophage polarization with the rHCI and rHCIII treatment compared to the control treatment [29].

As demonstrated in the above studies, the cardioprotective potential of collagen-based monotherapy following MI appears to be driven by a tandem mechanical and biological mechanism.

### 4.2. Fibrin

Fibrin is a polypeptide molecule generated from its precursor fibrinogen, familiar for its crucial function in hemostasis [46]. In the infarcted heart, fibrin is a component of the plasma-derived provisional matrix that facilitates the initial inflammatory response, as described previously [7,13]. Like collagen, fibrin has been used primarily as a scaffold or patch component in IHD therapy models [47,48]; also, like collagen, there have been attempts at its use as a monotherapy as well.

In keeping with its provisional matrix role in facilitating the release of growth factors that promote wound healing and tissue repair, in vitro models of fibrin support its pro-angiogenic potential [49]. In an acute LAD ischemia–reperfusion rat model, Christman et al. injected myoblasts seeded in fibrin-based scaffolds after MI; the effects of fibrin glue alone were also explored in this study through a histopathological assessment of infarcted areas [30]: compared to saline-injected controls, the fibrin treatment reduced the infarct scar size (26.5 ± 2.2% versus 19.7 ± 3.8%, respectively). Additionally, the treatment with fibrin glue in ischemic tissue resulted in neovascularization, as evidenced by the presence of a higher arteriolar density of actin measured within the infarct in the fibrin treatment group (13 ± 1 arterioles/mm^2^) compared to the control group (10 ± 2 arterioles/mm^2^). Another study examined the use of fibrin or alginate to thicken LV walls 5 weeks post-MI in rats [31]. In the fibrin treatment groups, echocardiography revealed improved fractional shortening (FS) (29.3% to 40.8%), increased anterior systolic (1.90 mm ± 0.54 mm to 3.24 mm ± 0.74 mm) and diastolic wall thickness, and decreased systolic (0.51 cm ± 0.08 cm to 0.38 cm ± 0.05 cm) and diastolic (0.72 cm ± 0.05 cm to 0.65 cm ± 0.06 cm) left ventricular internal dimension (LVID) 2 days post-treatment compared to baseline measurements before treatment. The treatment with alginate produced similar results 2 days post-treatment, while saline controls showed no significant improvement. 5 weeks post-treatment, echocardiography was repeated and revealed the deterioration of FS and systolic volume in the fibrin group, while the rest of the measured parameters were not significantly changed. The alginate-based treatment, by contrast, demonstrated better outcomes 5 weeks post-treatment in all aforementioned parameters. The infarct size in the fibrin treatment group was significantly reduced compared to the control group (12% ± 2% versus 16% ± 3%, respectively) [31], and the assessment of neovascularization revealed increased arteriolar density in the alginate (22 ± 8/mm^2^) and fibrin (24 ± 7/mm^2^) groups compared to the control group (14 ± 1/mm^2^). Neither fibrin nor alginate appeared to elicit an inflammatory response, as there were no significant differences in the presence of macrophages in these groups.

These promising preliminary results warrant additional experimental evaluation and validation in clinically relevant models of human acute and chronic ischemia.

### 4.3. Hyaluronic Acid

Hyaluronic acid (HA) is a long, unbranched polysaccharide found in the ECM across multiple different organ systems and tissues [50]. As with collagen and fibrin, HA-based therapy in MI has been explored in isolation and in addition to other components. These include HA in combination with cells (such as bone marrow-derived mononuclear cells or mesenchymal stem cells) or other polymers (such as chitosan, silk, or gelatin) [51]. HA is a crucial mediator of angiogenesis and wound repair, rendering it an attractive target in developing therapeutic models for IHD [52,53]. In post-embryonic zebrafish, epicardial cells expressing the *hapln1* gene are responsible for modulating HA production, as well as the expansion and regeneration of CMs; it is thus not surprising that the disruption of this gene resulted in the inhibition of CM proliferation. This indicates that HA signaling pathways within the ECM may play a role in cardiac cellular maturation and proliferation [54]. This is further corroborated by a similar knockout study of the *hapln1* gene in mice, which yielded reduced myocardial proliferation, among many other cardiac defects [55].

Several in vivo models have investigated the role of hyaluronic acid as a monotherapy for IHD. Abdalla et al. used HA-based hydrogels and injected them in a rat MI model [32]. An echocardiographic assessment of mean FS and mean LVEF was then performed and demonstrated the deterioration of both metrics from postoperative weeks 1–4 in the control group, while the HA injection produced significant improvements, with FS increasing from 23.6% to 33.2%, and mean LVEF from 43.1% to 55.6%. The histopathological assessment revealed neovascularization in the HA hydrogel group, but no evidence of neovascularization was observed in the control; and the scar area, as assessed through collagen deposition, was 22.6% smaller in the HA hydrogel group. These findings suggest that HA-based therapy may mitigate post-infarct adverse remodeling. Another study found similar results using an injectable HA hydrogel in infarcted rats [33]. A functional assessment using a pressure–volume catheter system revealed improvements in EF, SV, and diastolic function in the HA hydrogel group compared to control animals at 4 weeks post-injection. In fact, all functional parameters that demonstrated improvement were statistically different when compared to the sham surgery group at 4 weeks, suggesting a marked myocardial functional recovery through therapy. As in the previous study, the infarct size was smaller in the hydrogel group compared to the MI control, and a fibrosis assay revealed little to no fibrous tissue formation in the border zone of the hydrogel group when compared to MI control. Higher capillary and arteriolar densities were seen in the hydrogel group (19 ± 3.33 and 571.52 ± 155.71, respectively) compared to the MI control (12.5 ± 3.59 and 392.13 ± 120.69), demonstrating a pro-angiogenic effect. In addition, the group treated with HA showed evidence of improved ventricular wall thickness, and reduced apoptosis.

All these findings suggest that the HA-based treatment alters the ECM microenvironment in a manner that may prompt improved myocardial repair.

### 4.4. Periostin

Periostin is a matricellular protein that binds to many other ECM proteins and has roles in the regulation of signal transduction and tissue remodeling [56]. In the cardiac ECM, periostin is seen after the onset of MI and is involved in the remodeling process [56,57,58]. The importance of periostin in cardiac repair was demonstrated in a mouse knockout model in which, in wild type *periostin* +/+ mice, the expression of periostin was absent in non-infarcted cardiac tissue but was induced after MI [58]. In periostin −/− mice, higher mortality by cardiac rupture was observed when compared to periostin *+*/*+* mice or periostin +/− mice after MI. This, in conjunction with the abnormal collagen formation and lower number of cardiac fibroblasts also noted in periostin −/− mice compared to periostin +/+ mice, suggests that periostin is a critical component in post-MI wound healing. Additionally, periostin −/− mice that did not undergo MI did not display any abnormalities within their ECM or overall cardiac function, implying that the function of periostin is linked to post-pathologic recovery [59]. This role has been corroborated by other studies and appears to be meditated by the induction of mitogenesis in myocytes [60,61].

In one in vitro study, periostin induced the cycling of differentiated rat CMs, with cytokinesis and mononucleated CMs detected in cells treated with recombinant human periostin [62]. This was corroborated in the same study with an in vivo model using infarcted adult male rats, where the cycling and division of differentiated CMs was detected in 0.6% to 1% of myocytes at the infarct borders after myocardial periostin administration through a biodegradable gelfoam patch. Echocardiographic measurements recorded at 1 and 12 weeks post-treatment indicated improved EF (53 to 66%) and FS (25 to 33%). Evidence of attenuated remodeling and the functional consequences thereof were seen as a smaller left ventricular end-diastolic dimension in the periostin-treated group, accompanied by P-V loop measurements, indicating a steeper end-systolic pressure volume relationship (ESPVR) slope, higher preload, higher maximum rate of ventricular pressure rise, and higher maximum ventricular elastance. Similar findings were seen in another study that examined cardiac explant-derived progenitor cells (CPCs) that secreted exosomes containing short isoforms of periostin [63]. The exosome delivery resulted in the proliferation of adult cardiomyocytes in a model of left coronary ligation in adult rats. Re-entry to the cell cycle was seen uniquely with the short isomers of periostin, and not with full-length recombinant human periostin.

Periostin delivery has also been investigated in a large animal model: in swine subjected to LAD ligation, the introduction of periostin and gelfoam into the pericardial space improved EF (from 31.4 to 40.7%) and peak ejection rate (151 ± 11 mL/s versus 98 ± 20 mL/s in MI control) over a 3-month period [36]. Histologically, myocardial strips accounting for 7.1 ± 2.8% of infarct volume were observed within the middle of the scar; and structurally, animals treated with periostin had an infarct volume 27% smaller compared to control animals. This is consistent with earlier studies suggestive of the regenerative potential of periostin. Additionally, however, an increase in fibrosis was noted at remote regions. The authors hypothesized that this increase in fibrosis was due to the proliferation of fibroblasts seen in vitro in the same study. This is consistent with the aforementioned function of periostin in post-MI repair.

### 4.5. Agrin

Agrin is a heparan sulfate-based proteoglycan that is normally present in the ECM and involved with neuromuscular junction formation [64]. In the myocardium, the function of agrin appears to involve CM proliferation. This is supported by a study showing elevated levels of agrin present in P1 neonatal mouse cardiomyocytes as compared with P7 neonatal cardiomyocytes, which coincided with a decreased cardiac regenerative capacity in the latter [38]. In an in vitro arm from the same study, agrin induced the proliferation of both P1 and P7 cardiomyocytes and delayed their maturation; the same findings were seen in human-induced pluripotent stem cell cardiomyocytes in 2D and 3D culture systems. Additionally, a knockout study of agrin in P1 and P7 mice resulted in reduced cardiomyocyte proliferation and a mild disturbance in FS at one month that did not persist at later measurements, indicating that agrin is necessary for neonatal cardiac regeneration. To test agrin in the context of pathology, an intramyocardial administration of a single dose of recombinant agrin was performed in mice after LAD ligation. The treated mice demonstrated evidence of CM cell cycle re-entry, along with improved FS, EF, and wall thickness. A follow-up study using LAD ischemia–reperfusion in swine was performed, and concordantly demonstrated improved EF and left ventricular end-diastolic pressure (LVEDP), as well as reduced scarring and remodeling [37]. Agrin appears to function in a pleiotropic manner in this setting, stimulating CM proliferation, angiogenesis, and immunomodulation: in both studies, all these effects were seen after the administration of one or two local doses of 33 µg/kg of recombinant agrin.

Other studies have attempted to elucidate the molecular mechanism by which agrin promotes cardiac proliferation and dedifferentiation, and have yielded evidence of action through Lrp-4 and α-dystroglycan (DAG1) receptors [65,66]; the latter was expressed in P1 and P7 hearts in the study just described [38], in which agrin–DAG1 interaction induced ERK activation in cell culture, and the blockage of the ERK pathway using MEK inhibitors or agrin–DAG1 interaction through anti-DAG1 antibodies prevented the proliferation of P7 CMs. The mechanism may also involve the disassembly of the cytoskeletal dystrophin–glycoprotein complex and translocation of the YAP molecule, which is the effector protein of the Hippo signaling pathway [67]: binding of DAG1 to DGC in P7 CMs reduced the stability of the complex, and subsequently led to myofibril breakdown and molecular findings consistent with instability of the cytoskeleton and CM dedifferentiation. Following the treatment of CMs with agrin, the YAP molecule dissociated from the DGC and was found within the nucleus, indicating that agrin acts through YAP signaling to promote CM proliferation. This was verified by the co-administration of agrin and verteporfin, an inhibitor of YAP transcription, which resulted in the failure of CM proliferation [38].

### 4.6. Laminins

Laminins (LNs) are large glycoproteins consisting of α, β, and γ chains that are major components of basement membranes that function in maintaining the stability and integrity of the basement membrane as well as acting as a medium for cellular interactions [68]. Because of their heterotrimeric structure, several laminin isoforms can be assembled depending on their constituent heterotrimeric chains, such as laminin-221, -511, and -521 [69,70]. One study measuring laminin levels in patients following acute MI revealed that laminin levels were significantly higher compared to patients with stable coronary artery disease (CAD) (36.5 vs. 23.9 ng/mL) or without CAD (36.5 vs. 24.6 ng/mL), suggesting that they play a role in the post-infarct remodeling process [71]. In rats following myocardial infarction, laminin deposition was noted, with a distribution that started at the periphery of the infarct border within granulation tissue, and gradually extended to the outer lesion of the central infarct zone, suggesting that LNs are implicated in the repair process [72]. Furthermore, higher LN levels may correlate with prognosis following acute MI as seen with one study which demonstrated a higher degree of major adverse cardiac events in patients with higher LN levels [73]. As such, the modulation of LN may represent an opportunity for treatment. Yap et al. demonstrated that seeding human embryonal stem cells (hESCs) with LN-221 and LN-521 drives their differentiation towards a CM lineage, generating cardiovascular progenitor cells [74]. Sougawa et al. demonstrated that the epicardial delivery of LN-511-immersed collagen in acute MI rat models resulted in decreased infarct size and a maintained EF compared to the control group 8 weeks post-MI [39]. Furthermore, LM-511 induced angiogenic cytokine production and increased mature capillary density as denoted by the presence of isolectin B4 (ILB4) and smooth muscle actin (SMA) double positive cells compared to the control group (67.3 ± 7.6 cells/mm^2^ vs. 34.9 ± 2.2 cells/mm^2^). Similarly, in a separate study, the coadministration of LN-221 and a prostacyclin agonist on a rat LAD acute MI model using epicardial LN-221- and prostacyclin-immersed collagen sheets resulted in preserved EF and FS 4 weeks post-MI compared to controls [75]. Additionally, reduced infarct size, suppressed cell apoptosis, increased angiogenic and chemotactic cytokine production, and increased capillary maturation, as indicated by a higher density of ILB4 and SMA double positive cells (240.6  ±  15.5 cells/mm^2^ vs. control 57.9  ±  8.4 cells/mm^2^), were also noted.

### 4.7. Other ECM Components

Fibronectin (FN) is a glycoprotein that normally assumes a dimeric structure within the ECM, with each monomer consisting of three repeating units referred to as types I, II, and III. Two types of type III units exist: EDA- and EDB-containing isoforms. FN is further subcategorized by its source, as it may be derived either from plasma (secreted primarily by hepatocytes) or from cells [76]. The role of FN in cardiac repair was explored in a zebrafish knockout model [77]. As shown through the quantification of proliferating CMs in wild-type and transgenic homozygous fish with a loss of FN function, FN inhibition did not directly inhibit CM proliferation. A cellular density analysis in this model revealed that CMs in homozygous mutant *fn1* hearts, rather than migrating toward the infarct, aggregated at its borders, prompting the authors to hypothesize a larger role for FN in migration than in proliferation.

FN has also been associated with cardiac fibrosis and remodeling. In another study comparing EDA FN −/− knockout mice and wild type mice, it was noted that knockout resulted in the attenuation of fibrotic remodeling, preserved cardiac structure and geometry, improved contractility, lower LVEDP, and decreased monocyte recruitment within the scar tissue and remote myocardium [78]. These findings suggest the therapeutic potential for FN-target therapies; the existence of different isoforms of FN from different origins that may exert unique effects on cellular and non-cellular components within the cardiac ECM warrant caution and additional investigation.

One study conducted to investigate the role of cardiac fibroblasts on cardiomyocyte cytokinesis identified the ECM proteins SLIT2 and nephronectin (NPNT) in embryonic mouse hearts [79]. SLIT2 is a glycoprotein that acts as a repellant for axon migration during brain development [80], while NPNT is a glycosylated ECM protein that functions in the organogenesis of many different tissue types [81]. SLIT2 and NPNT produce mitotic rounding and cytokinesis in postnatal CMs, both in vitro and in vivo, and NPNT has been shown to promote migration and tube formation in human umbilical vein endothelial cells (HUVECs) through the EFGR/JAK2/STAT3 pathway [81]. This was confirmed by the impaired migration of HUVECs after treatment with Gefitinib, which blocks EGFR. The overexpression of NPNT in a mouse MI model yielded increased capillary and arteriolar density; a reduced infarct size; improved EF and FS; and a decreased left ventricular internal diameter in systole and diastole. This therapeutic promise warrants follow-up in further studies.

## 5. Tissue-Derived ECM

Given that the ECM consists of many cellular and noncellular components, the use of individual agents risks the loss of any therapeutic efficacy that may arise from the native interaction of these components in the tissue microenvironment. The use of a decellularized ECM (dECM) from various tissues and animals may, through the preservation of the original microenvironment and structure, resolve this potential problem. The administration of a dECM can be achieved through multiple means, including solid dECM patches, injectable dECM hydrogels, and injectable dECM microparticles [82]. Tissue-derived dECMs can be extracted from both animal and human sources for the treatment of IHD (Figure 4). The main sources of the ECM are either derived from animals or humans. The most widely used ECM tissue for IHD treatment is the porcine myocardial ECM [82,83]. Other tissues utilized to extract a dECM include the pericardial [84], urinary bladder [85], placental [86], and bone marrow ECM [87].

Briefly, the decellularization process involves placing cut-out pieces of selected tissue in a detergent (such as sodium dodecyl sulfate) to remove all cellular components while preserving the three-dimensional nature of the supporting ECM [88]. Figure 4 demonstrates the basic decellularization process of the porcine myocardial ECM. The ECM can be processed into an injectable hydrogel via lyophilization or cut into slices to form solid patches of decellularized ECM [89,90].

### 5.1. Animal-Based Decellularized ECMs

The most widely investigated ECM-based intervention for IHD as a monotherapy is the animal-based dECM. Animals that have been utilized as a source include pigs [89], rats [91], zebrafish [92], and sheep [93].

Of these, the porcine-derived myocardial ECM (PMM) is one of the most common [88,94,95,96,97,98]. Singelyn et al. conducted a study to assess the in vitro and in vivo effects of a dECM from pig hearts [88]. The biocompatibility of this source with neonatal rat CMs was tested, demonstrating that CMs remained viable. Migration studies assessed the ability of the dECM to support neovascularization, revealing that the matrix supported rat aortic smooth muscle cell (RASMC) and human coronary artery endothelial cell (HCAEC) migration. In vivo viability testing in male rats entailed dECM injections into the ventricular wall and found that the matrix induced a significant increase in arteriolar density 11 days post-injection. Another study also assessed the regenerative capacity of PMMs with CPCs isolated from rats. [94]. The PMM yielded an increased expression of early CM markers including NK2 homeobox 5, alpha-myosin heavy chain, and troponin C when compared to CPCs seeded in collagen as a control, suggesting that that the matrix possessed the capacity to affect differentiation. This was corroborated using Western blot analyses that demonstrated an increased expression of GATA-4 and NK2 homeobox 5 in the ECM cell group. CPCs seeded in the ECM also demonstrated a 35% increase in proliferation when compared to the control groups, along with reduced apoptosis when the CPCs were serum-deprived. All these findings were confirmed to be specific for the cardiac-based ECM by comparison with cells seeded in an adipose-derived ECM, which yielded similar findings to the control groups. Similar findings corroborated these results with human CPCs in a separate study [99].

Several studies have also explored the persistence of these findings in animal models [89,100,101,102,103]. Wassenaar et al. used a porcine ECM injection in an ischemia–reperfusion model, and showed an altered inflammatory profile of infiltrating macrophages, increased arteriole and capillary density along with increased pro-angiogenic growth factor expression, decreased CM apoptosis, increased CPCs, and attenuated fibrosis and hypertrophy in the treated group [100]. The increased expression of CD68 and MMP-12 found in this study, typically a sign of increased macrophage infiltration, was discordant with IHC staining for CD68, which revealed similar cell counts between the matrix and control groups. The authors proposed that the increased transcription of CD68 and MMP-12 represented a phenotypic shift in extant macrophages to reconcile this discrepancy, but no change in M1 vs. M2 macrophage predominance was detected [100,104]. In accordance with similar in vitro studies, studies on matrix-treated rats revealed evidence of neovascularization along with increased stem cell recruitment; a decrease in the negative regulator angiopoietin-2 coupled with increases in positive regulators fibroblast growth factors (FGFs) and vascular endothelial growth factors (VEGF) A and B may account for this pro-angiogenic shift. The increased expression of early cardiac and maturation markers GATA4, NK2 homeobox 5, MEF2d, myocardin, Tbx5, and Tbx20, along with an increased infiltration of c-KIT+ cells, was also observed. Tissue morphologic changes were also present, with hypertrophic CMs and reduced interstitial fibrosis seen in the treatment group [100]. Another study, in this case using an intramyocardial injection of ECM 2 weeks after MI induction in pigs, was conducted to investigate the efficacy of the pECM in a large animal model [89]. Echocardiographic data were compared to control groups across a 3-month period, revealing improvements in EF (43.3 ± 7.6 in control versus 73.7 ± 5.3% with ECM), LVESV (31.9 ± 5.0 in control; versus 8.1 ± 1.6 with ECM), and LVEDV (55.8 ± 2.8 in control; versus 33.5 ± 6.2 with ECM). Additionally, compatibility testing with human blood revealed no change in prothrombin time or activated partial thromboplastin time, no alteration in platelet function, and normal platelet activation.

The porcine myocardial ECM is the only tissue-derived animal-based ECM that is undergoing a phase I clinical trial (NCT02305602), under the brand name VentriGel [105]. The primary endpoint is the safety of VentriGel over a 6-month period after MI, and secondary endpoints assess efficacy using ESV, EDV, EF, and scar size measurements. No published results currently exist for this trial.

Other forms of the porcine-based ECM have been investigated for IHD. The pericardial porcine matrix (PPM) provides on example of this; a study that compared the PPM to the human pericardial matrix, collagen, and fetal bovine serum found that rat epicardial cells, RASMCs, and HCAECs all preferentially migrated towards the PPM [84]. Additionally, a PPM injection into the left ventricle of non-infarcted adult male rats was found to increase arteriolar density and potential stem cell infiltration, as indicated by the presence of c-KIT+ cells. In another study, porcine urinary bladder (UBM) patches, used as a scaffold for cardiac repair, were employed in a pig MI model [85]. Patches were applied 6–8 weeks post-MI and compared with synthetic expanded polytetrafluoroethylene (ePTFE) patches. After 3 months, the UBM patches appeared superior, with histologic and immunocytochemical studies revealing myofibroblasts, α-sarcomeric actin-positive, and α-SMA-positive cells within the contractile collagen-rich tissue region that formed within the infarct in this group. The group treated with ePTFE, by contrast, was characterized by calcification, necrosis, foreign-body giant cell reaction, and abscesses. Flow cytometry measurements at 1 and 3 months comparing normal myocardium, UBM-patched regions, and ePTFE-patched regions revealed higher infiltrating CD45+ inflammatory cells in ePTFE and UBM regions compared with the normal myocardium, along with higher α-SMA-positive cells in UBM-regions compared to ePTFE regions.

These findings indicate the strong potential for cardiac reparative efficacy using the pECM; results in patients are eagerly anticipated. Many other animal ECM sources—such as mice, zebrafish, and rats—have also been used, but in most cases in combination with other interventions, including stem cells, growth factors, drugs, and other synthetic material, rather than as a monotherapy [82].

### 5.2. Human ECM

The use of the human cardiac ECM for the treatment of IHD confers advantages that are difficult to obtain from xenogenous alternatives, most prominently including the reduced risk of immunogenic adverse effects. One study performed a comparison between hydrogels of the human cadaveric myocardial matrix (HMM) and PMM [106], and found that the properties between human and porcine myocardial matrices are similar, but with the important distinction that the sulfated glycosaminoglycan (sGAG) content of the HMM was significantly lower than that seen in the PMM. The is likely due to the increased age of the cadaveric samples; aging hearts are known to show increased fibrosis and adipose tissue deposition [107]. Moreover, it has been previously established that the age of an animal affects the properties of small intestinal submucosal ECM scaffolds; it thus stands to reason that similar effects would arise with use of the HMM [108]. Mechanistically, sGAGs within the ECM bind and sequester growth factors that are likely necessary for the therapeutic efficacy of the ECM, raising the possibility that impaired efficacy may result from the use of these aged sources [109]. The decellularization of the HMM was also reported to be more cumbersome when compared with the PMM: applying the protocol established for the latter to the HMM resulted in substantial remnant DNA and lipid content. Thus, a longer decellularization period in sodium dodecyl sulfate and additional steps to remove DNA and lipids were needed to achieve complete decellularization [106]. These differences may correlate with a decrement in efficacy, as an analysis of the HMM in the same study demonstrated that the PMM induced a higher expression of the early development cardiac marker NK2 homeobox 5 in human fetal cardiac progenitor cells. The evidence was mixed, however, as the HMM also appeared to cause the increased proliferation of RASMCs and HCAECs when compared with the PMM. In another study that used myocardial ECM from patients with non-ischemic dilated cardiomyopathy, the authors tested effects on stem cell differentiation, using murine embryonic stem cells (ESCs) and induced pluripotent stem cells (iPSCs) [110]. In this case, the ECM drove stem cells towards a cardiomyocyte phenotype, as demonstrated by a significant increase in the expression of cardiac alpha myosin heavy polypeptide 6 (myh6), cardiac troponin T2 (Tnnt2), and NK2 homeobox 5, as well as the detection of cardiac troponin T and cardiac myosin heavy chain genes on IHC in cells treated with this form of HMM. Other forms of non-cardiac ECM (Matrigel^®^ and Geltrex^®^) used in the study alongside the negative control did not manifest the same expressional pattern.

The human pericardial-derived ECM has also been studied and is of interest given its potential for autologous harvest, thus avoiding both potential xenogenic and allogenic adverse sequelae. In two studies, Seif-Naraghi et al. obtained pericardial samples from of patients undergoing cardiothoracic surgeries [84,111], and subsequently showed that an injection of the human pericardial ECM into healthy rats produced increased arteriolar density at 2 weeks when compared to baseline levels [84]. C-KIT+ cells were also found at the injection sites, raising the prospect of the induction of cardiac regeneration by the recruitment of stem cells. Further studies on MI-based preclinical models are needed to further evaluate the potential of the human pericardial ECM through the study of functional parameters, angiogenesis, stem cell recruitment, and long-term therapeutic outcomes.

In one study, a human placental ECM (hpECM) hydrogel was studied, and was shown to support the growth of iPSC-derived CMs more rapidly than those seeded on fibronectin, gelatin, Matrigel^®^, and tissue culture plastic [86]. With the hpECM treatment, cells attached and exhibited synchronous contractions 24–30 h after seeding—this effect was not observed in cells seeded in the other substrates. An additional experiment performed on a rat MI model revealed a reduced infarct size compared to saline-injected controls and restored the electrical synchronization of the infarcted LV; no increased risk of arrythmia or other conductive abnormalities were reported.

In summary, human-based ECMs may offer patient-specific treatments for IHD and will likely serve as the focus of intense investigation in coming years. Of particular interest will be the prospect of drawing on the potential of accessible sources, such as the bone marrow. Currently studied ECM models and outcomes are delineated below in Table 2.

## 6. Bioengineered ECM

While the tissue-derived ECM maintains the complex nature of ECM signaling, it remains subject to many limitations, including the variation in ECM constitution that inevitably arises following the extraction of the decellularized ECM from multiple sources. [111]. Furthermore, the properties of the ECM differ with aging [124]; this difference and the associated therapeutic decrement was demonstrated in a study that compared a fetal cardiac ECM to a neonatal and adult cardiac ECM from rats and revealed a larger cardiomyocyte expansion and increased adhesion in cells seeded in the fetal cardiac ECM-coated surfaces, as compared to those seeded in adult and neonatal ECM-coated surfaces [91].

Given these findings, it may be necessary, in the interest of minimizing therapeutic discrepancies, to standardize age in the acquisition of the tissue-derived ECM. Alternatively, by providing a standardized, customizable ECM tailored to the needs of patients on an individual level, bioengineering a three-dimensional (3D) ECM is a strategy that bypasses this issue entirely (Figure 5). To faithfully represent the native architecture and metabolic activity of the tissue-derived ECM, this strategy requires a sophisticated, 3D synthetic methodology: one available method involves seeding cells on scaffolds and allowing these to generate an ECM. The tendency of scaffolds to modulate the differentiation of cells through interaction systems between scaffold stiffness and cell forces exerted on the scaffold must be borne in mind during use [125,126,127]. Additionally, it has been observed that new tissue generated in this manner does not usually synchronize with scaffold degradation, rendering remodeling difficult [127].

Self-assembling, scaffold-free microtissue formation is a novel bioengineering method that may obviate some of these issues [127,128,129,130,131,132]. Rather than affixing them to a scaffold, cells cultivated with this method adhere to one another and self-assemble into microtissues within molds. Altering the shape of the molds allows investigators to alter the shape of embedded microtissue and generate cell-derived tension, which aligns cells in a specific orientation and causes the ECM they form to align in concert [130]. In addition, by changing the cohort of cells used, different forms of the ECM with differing mechanical and biological properties can be produced. The mechanical properties of the microtissue produced using this method has been shown to align with those of natural tissue; thus, it may be possible to achieve a significant reduction in variability and the prevention of allogeneic and xenogeneic reactions, and a tailored patient-specific ECM can be formed to address IHD using this method [130]. Figure 5 demonstrates the basic principles of both methodologies of bioengineering a 3D ECM.

## 7. Current Issues and Future Potential of ECM Treatment

With each form of ECM monotherapy comes a unique set of challenges. Individual ECM components fail to recapitulate the three-dimensional structure of the native ECM, and interactions between components are lost with this approach. The tissue-derived ECM, on the other hand, suffers from variability arising from heterogeneous ages and species of origin, in addition to conferring the potential for adverse allogeneic or xenogeneic consequences, including the prospect of disease transmission [91,124]. Additionally, although the tissue-derived ECM can be broken down and molded into specific shapes, its architecture or composition cannot be fully modified [130]. The variability and lack of versatility from which this modality suffers may give rise to difficulty with the reproducibility of the results it attains. Finally, the bioengineered ECM may address the standardization and customization challenges posed by tissue-derived ECM, but methods to generate an ECM that faithfully mimics the properties of native tissue will require extensive preliminary testing prior to attempts to elucidate efficacy.

Currently, the porcine cECM is among the very few forms of ECM monotherapy that has been investigated for safety and efficacy in humans [89,133,134]. Of the individual components, collagen has been examined extensively for its effects and safety in IHD [44]. This cannot be said for the remainder of the ECM modalities. The PPM, for example, has been studied in the context of delivering hepatocyte growth factor (HGF), and in a separate study comparing its effect with HPM [84,135], but no other studies exist to corroborate the resultant findings. All other forms of ECM-based treatment, such as other animal sources (zebrafish, rat, goat, cow, rabbit, sheep, etc.), alternative tissue sources (bone marrow, liver, bladder, small intestines, etc.), ECM components (fibronectin, agrin, periostin, fibrin, nephronectin, etc.), and the bioengineered ECM, still await further evaluation. In addition to the phase I trial currently investigating the PMM mentioned previously, there has been another clinical trial that attempted to assess the efficacy of alginate for dilated cardiomyopathy (AUGMENT-HF), but was conducted in the setting of heart failure, rather than in IHD, and thus falls outside the scope of this review [136,137].

Another potential limitation of ECM monotherapy is that it targets processes localized to the myocardium. Given the many comorbidities—including diabetes, hypertension, and chronic kidney disease—that IHD often carries, this may be inadequate to alter the ultimate course of the disease and improve outcomes in patients [138]. Additionally, the interaction between ECM-based therapy and the established MI management paradigm in patients is poorly understood, and it is therefore unclear whether standard therapies, such as surgical and/or percutaneous revascularization, anticoagulants, antiplatelet drugs, beta receptor antagonists, nitroglycerin, and so on, are compatible with novel, ECM-based strategies [139,140]. There are multiple other dimensions along which the literature on ECM-based therapy is limited. One is temporal: almost all currently completed studies principally measured short-term outcomes, leaving residual questions regarding the persistence of cardiac functional, histopathological, and other results over longer periods [38,89,101]. In addition, most studies utilize models of acute ischemia to study the ECM, which has produced a deficiency of knowledge regarding the effect of these treatments in chronic myocardial ischemia.

## 8. Future Direction and Concluding Remarks

As is readily appreciated in the results of numerous investigations, the use of the ECM has immense potential to provide a much-needed novel therapeutic strategy for patients with IHD. Many forms of ECM monotherapy currently exist, some of which employ a single component, and others that entail the use of the whole ECM, either tissue-derived or biosynthetic, and either alone or in combination with other therapies. Many studies have demonstrated positive effects of these various forms of the ECM on cardiac functional parameters, stem cell recruitment and differentiation, angiogenesis, safety, and reproducibility. Given its novelty and the relative incipience of its incorporation into clinical studies, the use of the ECM to alter the course of IHD and other cardiovascular diseases warrants intensive investigation. As this work continues, it is the hope of the authors that the various deficiencies of the spectrum of ECM modalities—including the failure of individual ECM components to comprehend the complex network of interactions or the hierarchal structure of the ECM; the source-to-source variation and inability to fully customize or manipulate the tissue-derived ECM; and the interactions between cells and scaffolds in the bioengineered ECM—can be mitigated or even fully overcome. Novel methods involving scaffold-less microtissue ECM production are promising in this regard, but require further investigation to comprehensively determine the safety, efficacy, and translatability of these promising therapies into the clinical setting.

## Figures and Tables

**Figure 1 medsci-12-00008-f001:**
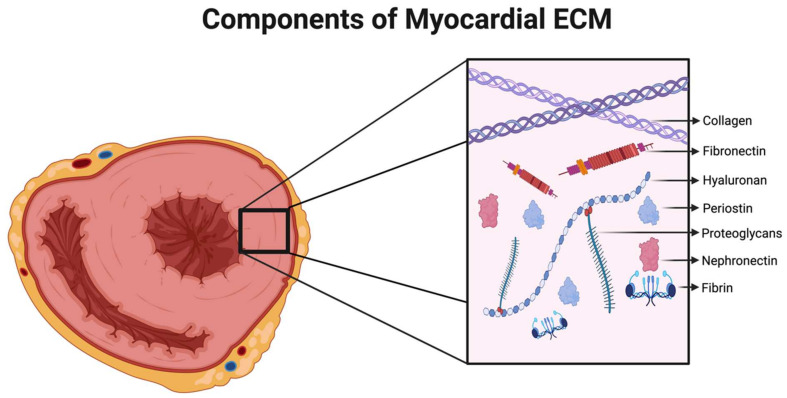
Components of ECM: Illustration summarizing the principal ECM components derived from myocardial ECM. Within the healthy myocardial ECM, components such as collagen, hyaluronan, and proteoglycans are present at varying levels. Following myocardial infarction (MI), components like fibrin, fibronectin, and periostin become prominent in the remodeling and repair processes. Additionally, during the early stages of fetal cardiac development, components such as agrin and nephronectin are observed in the ECM, playing key roles in the processes of proliferation and differentiation.

**Figure 2 medsci-12-00008-f002:**
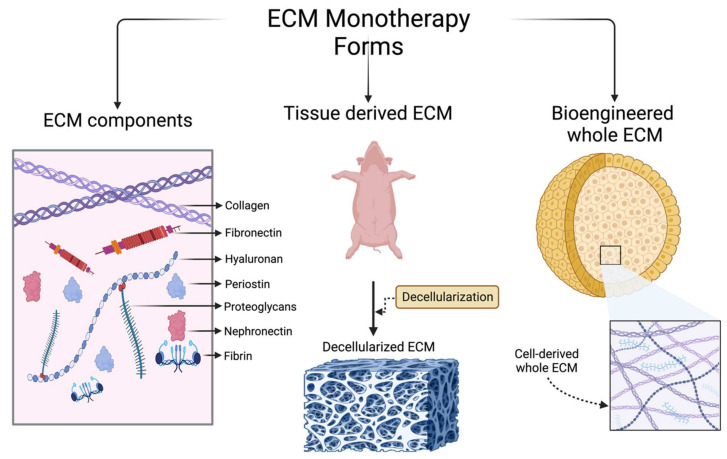
ECM in Ischemic Heart Disease (IHD)**:** Schematic summarizing several primary ECMs with potential for treatment of IHD. The direct intracardiac administration of individual ECM components, such as collagen, glycoproteins, and proteoglycans, can facilitate the repair of myocardial tissue). Another approach involves the extraction and decellularization of the ECM from the tissues to leverage its entirety, encompassing all components and their intricate interactions with both each other and cellular elements in myocardium, thereby promoting optimal tissue repair. The bioengineering method involves generation of ECM through modifiable techniques which can be employed to achieve standardized and individualized treatments for IHD.

**Figure 3 medsci-12-00008-f003:**
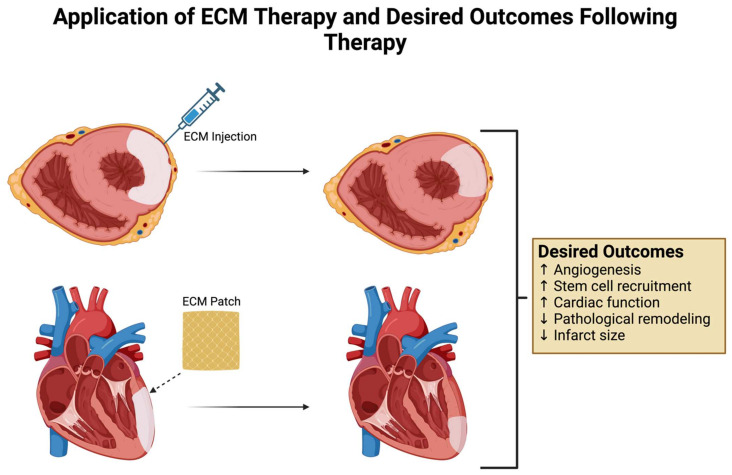
Application of ECM therapy and desired outcomes following therapy: Schematic highlighting two examples of applying ECM-based therapy. (**Upper panel**) Any form of ECM discussed can be loaded into an injection and applied directly into the infarcted myocardium. (**Lower panel**) ECM can be cut out into thin patches or loaded onto patches for delivery and applied directly over the infarcted surface of the heart. Following the application of ECM, several desirable outcomes including increased coronary angiogenesis leading to reduction in pathological remodeling, infarct size, and recovery of post-MI cardiac function are highlighted in the schematic.

**Figure 4 medsci-12-00008-f004:**
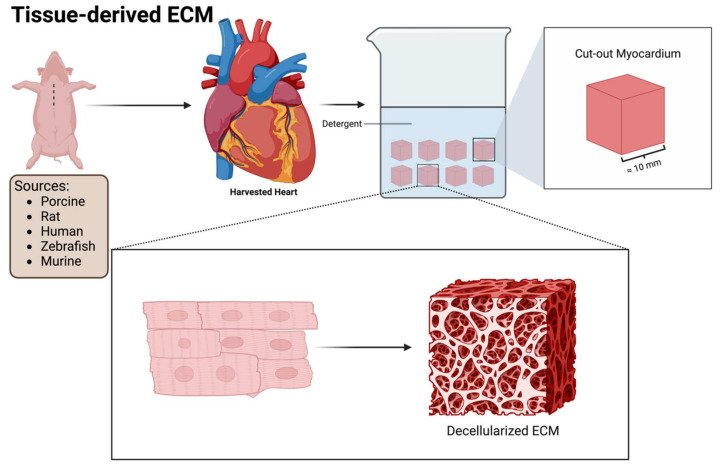
Tissue-derived dECM: illustration depicting the fundamental decellularization process employed in obtaining tissue-derived ECM from various sources. The chosen tissue undergoes subdivision into small pieces, subsequently immersed in a detergent solution. This process serves to eliminate all cellular components while preserving the structural integrity and constituents inherent in the ECM.

**Figure 5 medsci-12-00008-f005:**
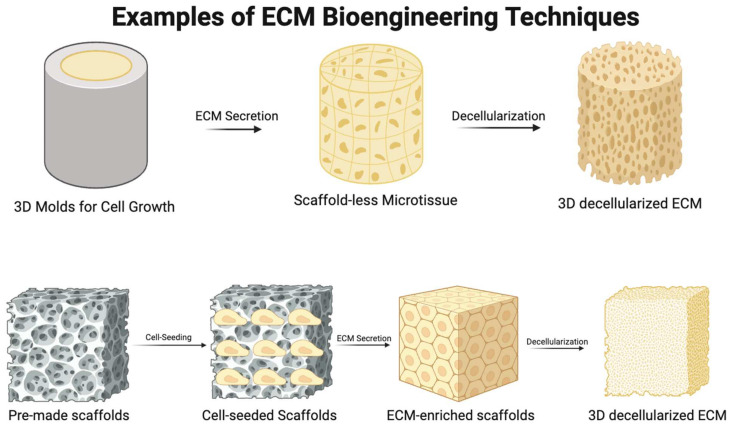
Different Methods for Bioengineering of ECM: Diagram illustrating two examples of bioengineering techniques for the generation of 3D bioengineered ECM. In the first approach, molds of various shapes can be seeded with cells, leading to the formation of scaffold-less microtissues (**upper panel**). These microtissues generate ECM with a modifiable architecture, utilizing forces derived from the cells themselves. Alternatively, another method involves seeding cells into pre-existing scaffolds to generate ECM based on the physical properties of the scaffold (**lower panel**).

**Table 1 medsci-12-00008-t001:** Summary of studies investigating different ECM components for IHD treatment.

Study	Component	Model	Major Findings
Dai et al., 2005 [28]	Type I collagen hydrogel	Rat MI	Treatment 1 week post-MI; increased SV, EF; decreased incidence of paradoxical systolic bulging
McLaughlin et al., 2019 [29]	Recombinant Type I and III collagen hydrogels	Mouse MI	Treatment 1 week post-MI; Type I superior to type III; improved EF, CO, and SV; reduced scar size, ESV, and cardiac dilation; increased capillary density and M2 macrophages within scar
Christman et al., 2004 [30]	Fibrin glue	Rat ischemia-reperfusion	Treatment 1 week post-MI; reduced infarct size and increased arteriolar density
Yu et al., 2009 [31]	Fibrin glue	Rat MI	Treatment 5 weeks post-MI; slowed deterioration of FS; maintained anterior systolic and diastolic wall thickness; decreased systolic and diastolic internal dimension; reduced infarct size; increased arteriolar presence
Abdalla et al., 2013 [32]	Hyaluronic acid hydrogel	Rat MI	Treatment at MI induction; increased FS, EF; fostered neovascularization; reduced collagen deposition
So et al., 2009 [33]	Hyaluronic acid hydrogel	Rat MI	Treatment 2 weeks post-MI; increased EF, SV, diastolic function; decreased infarct size; increased arteriolar and capillary density; increased ventricular wall thickness
Wang et al., 2019 [34]	Hyaluronic acid oligosaccharide injection	Mouse MI	IV injections every other day; reduced infarct size; increased LV wall thickness; improved EF, FS; augmented systolic and diastolic LVID; reduced apoptosis; increased microvessel density; increased macrophage migration and invasion alongside M2 polarization
Dorsey et al., 2015 [35]	Hyaluronic acid hydrogel	Pig MI	Treatment at MI induction; increased infarct wall thickness; improved EF; reduced cardiac dilation; increased infarct stiffness
Ladage et al., 2013 [36]	Periostin gelfoam	Pig MI	Treatment 2 days post-MI; improved contractility, EF, cardiac index; improved circumferential strain; reduced infarct size and increased peri-infarct vascularization; induced formation of myocardial strips within the infarct
Baehr et al., 2020 [37]	Recombinant agrin	Pig ischemia-reperfusion	Treatment at MI induction; improved EF; preserved EDP; decreased scarring and attenuated pathological remodeling
Bassat et al., 2017 [38]	Recombinant agrin	Mouse MI	Treatment at MI induction; improved FS, EF; retained wall thickness
Sougawa et al., 2019 [39]	Collagen-immersed LN-511	Rat MI	Treatment at MI induction; improved EF, decreased fibrosis, increased endothelial and smooth muscle cell recruitment

MI: Myocardial Infacrt; SV: stroke volume; EF: ejection fraction; CO: cardiac output; FS: fractional shortening; ESV: end-systolic volume; LVID: left ventricular internal dimension; EDP: end-diastolic pressure; LN-511: Laminin-511.

**Table 2 medsci-12-00008-t002:** Summary of studies investigating different forms of tissue-derived ECM for IHD treatment.

Study Authors	ECM Source	In Vitro Model	In Vivo Model	Major Findings
Seif-Naraghi et al., 2013 [89]	PMM		Pig MI	Treatment 2 weeks post-MI. Higher EF; improved GWMI; lower ESV and EDV; reduced infarct size and collagen
Wassenaar et al., 2016 [100]	PMM		Rat Ischemia-Reperfusion	Treatment 1-week post ischemic induction. Higher EF, LVPSP, improved relaxation and contractility, lower ESV; increased capillary and arteriolar density; decreased interstitial fibrosis; increased recruitment of CPCs
Singelyn et al., 2009 [88]	PMM	Neonatal rat CMs, HCAECs, RASMCs	Healthy Rats	CMs are viable in PMM gel; PMM promotes migration of HCAECs and RASMCs; increased arteriolar density 11 days post-injection within injected myocardial matrix compared to baseline measurements 4 h post-injection
French et al., 2012 [94]	PMM	Isolated rat CPCs		Increased gene expression of early CMs: NKx2.5, α-mhc, and troponin-C; increased protein expression of GATA-4 and NKx2.5; increased proliferation, survival, and adhesion compared to CPCs on COL
Duan et al., 2011 [112]	PMM + Collagen	hESCs		75% PMM and 25% COL mixture without supplemental GFs yielded best results: drove cardiac differentiation through higher expression of cTNT; promoted contractile function; fostered cardiac maturation
DeQuach et al., 2010 [97]	PMM	hESC-derived CMs		Cells cultured on gelatin used as control. PMM facilitated cardiac maturation: higher fold change of cell cluster area through late cardiac muscle-specific marker Titin M8, higher fold change of desmoplakin presence on the lateral ends of cells, and higher number of nuclei per cluster
Singelyn et al., 2012 [101]	PMM		Rat MI	Treatment 1 week post-MI with saline injection as control: increased area of viable myocardium within infarct zone; increased density of proliferative cells; PMM group preserved EF, ESV, and EDD compared to baseline measurements pre-injection, whereas saline-injected group showed deteriorating function on follow-up
Johnson et al., 2014 [106]	PMM and HMM	RASMCs, HCAECs, hfCPCs		Higher proliferation rates of RASMCs and HCAECs in HMM compared to PMM at 3 and 5 days; increased expression of NKx2.5 in hfCPCs seeded in PMM
Jeffords et al., 2015 [113]	PMM crosslinked with Genipin	hMSCs		Higher expression of early cardiovascular markers α-SMA and GATA-6 in cross-linked groups; decreased expression of early endothelial cell markers VE cadherin and CD31 compared to non-crosslinked PMM
Gaetani et al., 2016 [99]	PMM	hfCPCs and haCPCs		Increased expression of GATA-4, MLC2v, and VEGFR2 in hfCPCs at day 4; increased expression of the markers NKx2.5, MEF2c, VEGFR2, and CD31 in haCPCs at day 4; both cell types had higher metabolic activity in a hydrogen peroxide assay compared to control
Merna et al., 2015 [98]	PMM + LECM	HCFs and NHLFs		Elevated expression of β_3_ and β_4_ integrins along with lower expression of α-SMA in HCFs compared to NHLFs; inhibition of β_3_ in HCFs; increased α-SMA expression in HCFs cultured on PMM, suggesting that α-SMA variation among the two groups is due to differential expression of β_3_ integrin
Diaz et al., 2021 [114]	PMM		Rat MI	Treatment 8 weeks post-MI induction; maintenance of ESV and EDV in PMM group compared to deterioration in saline-injected group; decreased apical wall thickness in saline group vs. PMM group; modulation of chronic inflammation through decreased MMP2 and TIMP2 expression; downregulation of inflammatory cytokines
Wang et al., 2022 [115]	PMM microparticles + PMM hydrogel		Mouse MI	Treatment at time of MI induction; increased EF, FS, SV, and decreased ESD in both PMMs compared to MI control; increased small vessel density in microparticles compared to MI and hydrogel; increased wall thickness, decreased scar area, and decreased fibroblast activation in both PMM groups; decreased activated fibroblast density only in PMM microparticles; increased cell cycle activity in both groups
Sarig et al., 2016 [103]	PMM Patches		Rat MI	Treatment at time of MI induction (acute model) or 30 days post-MI (chronic model); decreased infarct size, LVID; increased LVWP and systolic IVS; increased FAC, FS, EF, and contractility; higher mean vessel density within patches in both models; recruitment and maturation of progenitors as indicated by time-dependent increase in infiltration of GATA-4+ (early marker) and MYLC+ (late marker) cells
Kong et al., 2023 [116]	PMM/GP hydrogel	HUVECs, MUVECs, BMDMs	Rat MI, Swine MI	Higher induction of M2 polarization compared to PMM or GP alone; promotion of EC migration and tube formation; induction of EC proliferation through crosstalk with M2 macrophages. Treatment at time of MI induction (rat model) and compared to PMM alone, GP alone, and saline control: improved EF, EDV, wall thickness, and decreased collagen deposition; higher vessel density within infarct and at border. Swine model treatment 7 days post-MI injection: improved EF, wall thickness, and decreased infarct area
Seif-Naraghi et al., 2010 [84]	PPM + HPM	RECs, RASMCs, HCAECs	Healthy Rat	Induction of migration of all three cell types preferentially towards PPM compared to HPM, FBS, and COL; increased vessel density 2 weeks post-injection compared to baseline measurements in both HPM and PPM
Wang et al., 2021 [117]	Fetal and adult PMM	P1 mouse ventricular explant, isolated rat ventricular FBs	P1 Mouse MI	Treatment at time of MI induction: decreased FB activation and collagen deposition with fetal PMM, but not adult PMM. Fetal PMM decreased FB activation and collagen deposition in explants; microenvironment stiffness promotes FB activation in explants and isolated ventricular FBs; fetal PMM can regulate FB activation through CAPG
Crapo et al., 2010 [118]	pSIS-ECM	P3 Mouse Mixed Cardiac Cells		pSIS-ECM compared to Matrigel; contraction rate of engineered tissue higher in pSIS-ECM; cells on pSIS-ECM displayed diffuse spread and uniform structure; troponin T protein expression higher in pSIS-ECM
Toeg et al., 2013 [119]	pSIS-ECM + CACs		Mouse MI	Treatment 1 week post-MI, compared to PBS-injected control; findings with and without addition of CACs were similar: increased EF and posterior LV wall thickness; reduced infarct size; increased arteriolar density; possible cardiomyogenesis at periinfarct regions through detection of CMs or CPCs expressing β-catenin localized to intercellular adherens junction and detection of GATA-4 expression in CMs or CPCs
Okada et al., 2010 [120]	pSIS-ECM		Mouse NOD-SCID MI	Treatment at MI induction. Two forms of pSIS used and compared to saline-injected control: pSIS-B and C (B has higher levels of basic fibroblast growth factor): decreased LVESA; increased FAC; decreased infarct size and scar area fraction; increased capillary density at peri-infarct and infarct zones in pSIS-B compared to control and pSIS-C
Ravi et al., 2012 [87]	pBM-ECM + methylcellulose	HUVECs	Rat MI	Treatment at MI induction; improved FS and reduced infarct size and fibrotic area; promoted HUVEC proliferation and adhesion; increased stem cell mobilization; augmented blood vessel density; reduced apoptotic cells
Williams et al., 2015 [121]	PD-RMM	P2-P3 rat cardiac cells		Increased CM density, population, and proliferation on PD-RMM with short solubilization period; PD-RMM solubilized for longer period supported sarcomere maturation; inverse relationship between sarcomere maturation and CM proliferation. FBS and PLL as positive and negative controls, respectively
Dai et al., 2013 [122]	RMM		Rat MI	Treatment 1 week post-MI, saline injection as control. Increased FS, EF; decreased LV systolic bulging; increased thickness of infarct wall; no significant differences in c-KIT+ stem cell recruitment; no signs of angiogenesis
Wang et al., 2019 [123]	nmECM	HUVEC	Mouse MI	Treatment at MI induction; improved FS, EF, FAC, ESA, and EDA; smaller scar size; reduced ventricular stiffening; promoted angiogenesis and endothelial cell activity
Chen et al., 2016 [92]	hzECM + nzECM	hCSCs + hHp	Mouse MI	Treatment at MI induction; improved FAC, EF, EDA, and ESA; preserved myocardial elasticity; pro-proliferative effect of zECM
Robinson et al., 2005 [85]	pUBM patches		Pig MI	Treatment at 6–8 weeks post-MI; presence of myofibroblasts, α-sarcomeric actin-positive, striated cells, and alpha smooth muscle actin after 3 months on histology
Francis et al., 2017 [86]	hpECM	iPSC derived CMs	Rat MI	Supports growth, proliferation, adhesion, and synchronization of iPSC CMs in vitro. With treatment at MI induction, reduced infarct size; restored electrical synchronization with no increased risk of conductive abnormalities

GWMI: global wall motion index; EDV: end-diastolic volume; LVPSP: left ventricular peak systolic pressure; FAC: fractional area change; IVS: inter-ventricular septum dimension; EDD: end-diastolic diameter; ESD: end-systolic diameter; ESA: end-systolic area; EDA: end-diastolic area; EC: endothelial cell; FBs: fibroblasts; CPCs: cardiac progenitor cells; CMs: cardiomyocytes; HCAECs: human coronary artery endothelial cells; RASMCs: rat aortic smooth muscle cells; HUVECs: human umbilical vein endothelial cells; hESCs: human embryonic stem cells; hfCPCs: human fetal cardiomyocyte progenitor cells; hMSCs: human mesenchymal stem cells; haCPCs: human adult cardiomyocyte progenitor cells; CACs: circulating angiogenic cells; NHLF: normal human lung fibroblasts; HCF: human cardiac fibroblasts; iPSCs: induced pluripotent stem cells; BMDMs: bone marrow-derived macrophages; RECs: rat epicardial cells; hCSCs: human cardiac stem cells; hHp: human heart perivascular mesenchymal stem/stromal cell-like precursors; PMM: porcine myocardial matrix; HMM: human myocardial matrix; LECM: lung extracellular matrix; PMM/GP: porcine myocardial matrix glycopeptide hybrid; PPM: porcine pericardial matrix; pSIS-ECM: porcine small intestinal submucosal extracellular matrix; pBM-ECM: porcine bone marrow extracellular matrix; RMM: Rat myocardial matrix; PD-RMM: partial digested RMM; nmECM: neonatal mouse ECM; nzECM: normal zebrafish ECM; hzECM: healing zebrafish ECM; pUBM: porcine urinary bladder matrix; hpECM: human placenta ECM; COL: collagen; cTnT: cardiac troponin T; FBS: fetal bovine serum; CAPG: macrophage capping protein; NOD-SCID: non-obese diabetic severe combined immunodeficiency mice.

## Data Availability

No new data were created or analyzed in this study. Data sharing is not applicable to this article.

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
