# Peer review of "The Current State of Extracellular Matrix Therapy for Ischemic Heart Disease"

_medsci, 2024, doi:10.3390/medsci12010008_

Round 1
Reviewer 1 Report
Comments and Suggestions for Authors
The manuscript deals with a very important topic in the field of regenerative medicine, the extracellular matrix is indeed a key issue in setting up a possible proper tissue regeneration. The authors focus on the possible role of ECM in the therapy of cardiac ischaemia and address the topic in an orderly and comprehensive manner. The text is well written and the bibliography extensive and up-to-date. The flourishes and tables are well done. Only some typographical details need to be corrected, in particular on page 6 line 169: CORRECT THE PARAGRAPH FORMATTING, on page 6 line 176: CHECK THE SENTENCE.
Reviewer 2 Report
Comments and Suggestions for Authors
This article entitled, “The Current State of Extracellular Matrix Therapy for Ischemic Heart Disease" by Hamsho et al., discussed the role of extracellular matrix in Ischemic Heart Disease. Within the myocardial, the extracellular matrix (ECM) functions as a framework composed of collagen, facilitating the transmission of signals for contraction, enabling paracrine signaling, and ensuring the maintenance of nutritional and immunologic balance. Myocardial infarction (MI) is a well-known example of this, as it involves changes in the extracellular matrix (ECM) that are typical during the several stages of recovery after an infarction. This study presents a comprehensive analysis of the existing literature about the application of extracellular matrix (ECM) as an innovative treatment for ischemic heart disease. The study specifically emphasizes biologically-derived models, including both the entire ECM and its individual components.
Altogether this is an important and timely article, this reviewer has certain suggestions that would help produce a more comprehensive overview of the topic:
Comments:
1, At least one additional Figure (illustration) may be provided as to highlight the summary or prospect of this study.
2, The English of manuscript can be polished (minor) and there are few typological errors.
3, Authors can add one paragraph for abbreviations.
4, Role of immune cells are also very important factor in cardiovascular disease, therefore I would suggest adding few citations to put comprehensive view of this topic (PMID: 36093172; PMID: 30354461; PMID: 33582093; PMID: 36337927; etc.).
Comments on the Quality of English LanguageMinor editing of English language required
Reviewer 3 Report
Comments and Suggestions for Authors
This is a nice revision of the current state-of-the-art in the field of ECM application for treating the heart after ischemic damage.
Comments on the Quality of English LanguageThe article reads easily, clearly and smoothly, except for the first chapter:
29-31 – punctuation marks
32-33 – extra a
38-40 – reorganize
37-39 – punctuation marks
41-42 – reorganize
Reviewer 4 Report
Comments and Suggestions for Authors
In this review manuscript, the authors begin with the explanation of extracellular matrix (ECM) components in the myocardial tissue, and describe many examples of ECM therapy for ischemic heart disease using individual ECM components, tissue-derived ECM, and bioengineered ECM. In addition, authors briefly describes the current issues and the potential of future ECM therapy. The manuscript is well written and well balanced, but Figure 4 seems to need some improvement. Overall, this manuscript will be very informative for researchers in the field of ECM therapy.
Comments
In the upper panel of Figure 4, you say that the microtissue is formed inside the mold, but the shape of the mold and the scaffold-less microtissue are different, so it is difficult to understand what kind of operation was performed. In the lower panel of Figure 4, the first two illustrations are two-dimensional and the latter two are three-dimensional. Does it have any meaning?
There is almost no description of basement membrane proteins such as laminin-221 and laminin-511. Why don’t the authors address the studies on these proteins in this review manuscript?
